# Coronaviral Main Protease Induces LPCAT3 Cleavage and Endoplasmic Reticulum (ER) Stress

**DOI:** 10.3390/v15081696

**Published:** 2023-08-05

**Authors:** Jia Wang, Meifang Zhang, Yanli Ding, Yuxi Lin, Yan Xue, Xiaohong Wang, Xin Wang

**Affiliations:** 1Key Laboratory of Marine Drugs, Chinese Ministry of Education, School of Medicine and Pharmacy, Ocean University of China, Qingdao 266003, China; wangjia990126@163.com (J.W.); zhangmeifang96@163.com (M.Z.); dingyl_h@163.com (Y.D.); linyx18@lzu.edu.cn (Y.L.); 2Key Laboratory of Tropical Biological Resources of Ministry of Education and One Health Institute, School of Pharmaceutical Sciences, Hainan University, Haikou 570228, China; 3Department of Physiology, School of Basic Medicine, Qingdao University, Qingdao 266071, China; xueyan0512@126.com; 4Shandong Foreign Trade Vocational College, Qingdao 266100, China; wxh_qhnu@163.com; 5Song Li’ Academician Workstation, School of Pharmaceutical Sciences, Hainan University, Sanya 572000, China

**Keywords:** Mpro, LPCAT3, gastrointestinal symptoms, ER stress, coronavirus diseases

## Abstract

Zoonotic coronaviruses infect mammals and birds, causing pulmonary and gastrointestinal infections. Some animal coronaviruses, such as the porcine epidemic diarrhea virus (PEDV) and transmissible gastroenteritis virus (TGEV), lead to severe diarrhea and animal deaths. Gastrointestinal symptoms were also found in COVID-19 and SARS patients. However, the pathogenesis of gastrointestinal symptoms in coronavirus diseases remains elusive. In this study, the main protease-induced LPCAT3 cleavage was monitored by exogenous gene expression and protease inhibitors, and the related regulation of gene expression was confirmed by qRT-PCR and gene knockdown. Interestingly, LPCAT3 plays an important role in lipid absorption in the intestines. The Mpro of coronaviruses causing diarrhea, such as PEDV and MERS-CoV, but not the Mpro of HCoV-OC43 and HCoV-HKU1, which could induce LPCAT3 cleavage. Mutagenesis analysis and inhibitor experiments indicated that LPCAT3 cleavage was independent of the catalytic activity of Mpro. Moreover, LPCAT3 cleavage in cells boosted CHOP and GRP78 expression, which were biomarkers of ER stress. Since LPCAT3 is critical for lipid absorption in the intestines and malabsorption may lead to diarrhea in coronavirus diseases, Mpro-induced LPCAT3 cleavage might trigger gastrointestinal symptoms during coronavirus infection.

## 1. Introduction

Coronaviruses are single-strand positive RNA viruses, which have been classified into four genera, namely the Alphacoronavirus (α-CoV), Betacoronavirus (β-CoV), Gammacoronavirus (γ-CoV) and Deltacoronavirus (δ-CoV) [1]. Coronaviruses infect mammals and birds [2]. Although coronaviruses are phylogenetically highly conserved, their infection causes distinct clinical symptoms [3]. Most coronaviruses cause no to mild symptoms, such as human coronavirus HKU1 (HCoV-HKU1) and human coronavirus OC43 (HCoV-OC43) [4]. However, some coronavirus diseases are severe, including porcine epidemic diarrhea virus (PEDV), porcine transmissible gastroenteritis virus (TGEV), infectious bronchitis virus (IBV), severe acute respiratory syndrome coronavirus 2 (SARS-CoV-2), and Middle East respiratory syndrome coronavirus (MERS-CoV) [5]. Interestingly, many animal coronaviruses cause severe gastrointestinal symptoms, such as decreased appetite, nausea, vomiting, and diarrhea, as well as possible abnormal liver function, but SARS-CoV-2, MERS-CoV, and SARS-CoV only cause moderate gastrointestinal symptoms even though they infect and replicate in the intestine [6,7]. Moreover, HCoV-HKU1 and HCoV-OC43 infect gastrointestinal tracts but mostly cause no or mild gastrointestinal symptoms [8]. It is interesting to explore the key factors of gastrointestinal symptoms for coronavirus diseases.

Coronaviruses encode 12–16 nonstructural proteins (NSPs) and similar numbers of structural and accessory proteins [9]. Approximately two-thirds of the coronavirus genome contains two large open reading frames (ORFs) comprising the replicase genes ORF1a and ORF1b, encoding the replicase polyproteins pp1a and pp1ab [10]. Another one-third of the coronavirus genome encodes structural and accessory proteins, which mainly include the spike (S) protein, the envelope (E) protein, the membrane (M) protein, and the nucleocapsid (N) protein [10]. As a complicated virus with a huge genome, coronaviral proteins contact many host genes, which facilitate virus replication and lead to clinical symptoms. Mpro is essential for the proteolysis of 11 different cleavage sites of SARS-CoV-2 polypeptides, which leads to the maturation of NSPs. In this case, the cleavage of viral polyproteins and the maturation of NSPs are essential for host infection and viral dissemination. The structures of approximately 20 Mpro complexes with different viral cleavage sequences have been determined, in which the Mpro homodimer has a peptide substrate bound to its active site. A systematic analysis identified 332 high-confidence interactions between 26 SARS-CoV-2 proteins and host proteins. These interactions might influence multiple biological processes, including lipoprotein metabolism, nuclear transportation, ribonucleoprotein complex biogenesis, membrane transportation, and mitochondrial matrix [11]. Notably, it revealed that SARS-CoV-2 Mpro interacted with HDAC2, and Mpro C145A mutation interacted with TRMT1 and GPX1 [11]. Later, the interactome of SARS-CoV-2 proteins and host proteins was pursued by other groups, providing details of SARS-CoV-2 targeted network communities, in which the interaction between Mpro and GPX1 was confirmed [12,13,14,15,16]. Interestingly, mutant Mpro interacted with different host proteins compared to wild-type Mpro, suggesting the interaction between Mpro and host proteins might be transient. Viral proteases might cleave associated proteins in a “hit-and-run” manner. Herein, it is interesting to identify the viral proteases and host protein interaction via other approaches. During infection, viral papain-like protease (PLpro) and the main protease (Mpro or 3CLpro) hydrolyze pp1a/pp1ab into 12–16 nonstructural proteins [17]. It becomes clear that viral proteases also interpret host proteins. SARS-CoV-2 PLpro removes ISG15 (interferon-stimulated gene 15) from ISGylated IRF3 (interferon regulatory factor 3) to inhibit innate immunity [18], while SARS-CoV-2 Mpro cleaves the signal transducer and activator of transcriptions (STATs), ring finger protein 20 (RNF20), and many others [11,18,19,20,21,22]. Mpro is relatively conserved with minor differences. However, the non-conserved motif causes different cleavages of substrates [23]. The selective cleavage of Mpro may contribute to the various clinical symptoms caused by coronavirus infection.

Glycerophospholipids play important structural and functional roles in cells, which are formed via the Kennedy pathway and are subsequently matured in the Lands’ cycle [24]. In Lands’ cycle, lysophospholipids are re-acylated by lysophospholipid acyltransferases (LPLATs) [25]. LPCATs are members of the LPLATs, which are widely distributed and show varying substrate preferences for acyl-CoAs and lysophospholipids, including LPCAT1–4 [26]. LPCAT3 is the main isoform in the liver and small intestine [26]. Intestinal LPCAT3 deficiency reduces lipid absorption, thus decreasing atherosclerogenic lipoproteins in circulation [27]. LPCAT3 knockdown in the liver exacerbates endoplasmic reticulum (ER) stress and inflammation [28]. Mostly, malabsorption and infection-induced inflammation in the intestines might lead to severe diarrhea [29]. Therefore, LPCAT3 deficiency may contribute to the gastrointestinal symptoms of coronavirus diseases.

Increasing evidence suggests the importance of ER stress response in coronavirus infection [30]. Viral infection may induce ER in multiple ways, such as rapid but inaccurate protein synthesis and protein maturation, the exhaust of amino acids, improper protein transportation, etc. ER stress caused by viral infection modulates various signaling pathways, leading to cell survival or cell death [31]. The glucose-regulated protein 78 (GRP78) has a nodal role in ER stress by interacting with three mediators: PKR-like ER kinase (PERK), activating transcription factor 6 (ATF6), and the ER transmembrane protein kinase/endoribonuclease (IRE1) [32]. Interestingly, membrane lipid saturation alteration can induce ER stress [32,33]. LPCAT3 deficiency alters membrane lipid saturation by decreasing unsaturated lipids in membranes, so it is not surprising to see that LPCAT3 knockdown promotes ER stress [28].

In this study, we report that Mpro induces LPCAT3 cleavage and ER stress. Interestingly, this cleavage was only found in viruses that caused gastrointestinal symptoms. Considering the critical role of LPCAT3 in lipid absorption and macrophage polarization, it is suggested that Mpro-induced LPCAT3 cleavage might lead to gastrointestinal symptoms in coronavirus diseases, especially severe diarrhea. LPCAT3 deficiency might be a common feature for highly virulent coronavirus infection.

## 2. Materials and Methods

### 2.1. Cell Culture

HEK293T, HeLa, HuH-7, HCT116, and MCF7 cells were obtained from the ATCC (American Type Culture Collection). IPEC-J2 cells were obtained from FenghBio (Changsha, China). HEK293T, HeLa, HuH-7, and MCF7 cells were cultured using Dulbecco’s modified Eagle medium (DMEM) (Cienry, Huzhou, China) (CR-12800). HCT116 cells were cultured using McCoy’s 5A medium (Cienry, Huzhou, China) (CR-16600). IPEC-J2 cells were cultured using RPMI-1640 medium (Cienry, Huzhou, China) (CR-31800). All culture mediums were supplemented with 10% fetal bovine serum (FBS) (Gibco, Carlsbad, USA), penicillin (100 IU/mL), and streptomycin (100 mg/mL) (HyClone, Logan, USA). Cells were maintained in an incubator containing 5% CO_2_ at 37 °C.

### 2.2. Recombinant Vectors and Transfection

Plasmid encoding SARS-CoV-2 (Wuhan strain) Mpro was stored in the lab. Full-length cDNA of MERS CoV Mpro, PEDV Mpro, HCoV-HKU1 Mpro, and HCoV-OC43 Mpro were synthesized (Sangon Biotech, Shanghai, China). LPCAT3 cDNA was purchased from Addgene. Target DNA sequences were amplified using high-fidelity 2 × PrimeStar max DNA polymerase (Takara, Dalian, China), and additional restriction enzyme sites as well as the additional tags. Amplified DNA fragments were digested by BamH I and XhoI, and then cloned into pcDNA3.1 by T4 ligase (NEB, Beijing, China). The SARS-CoV-2 Mpro (H41Y/C145S) mutant was generated using Mut Express II Fast Mutagenesis Kit V2 (Vazyme Biotech, Nanjing, China). For lentiviral vectors, full-length PEDV Mpro with a Flag-tag at the N-terminus was cloned into a pLVX-Puro vector. All constructions were validated by DNA sequencing, and the details of the plasmid sequence can be found in Appendix A.

Cells were transfected using lipofectamine 3000 (Thermo Fisher, Waltham, USA) following manufacturers’ protocols. To package lentiviruses, HEK293T cells were inoculated into the six-well plate. Moreover, 2 μg of pLVX-PEDV Mpro or empty pLVX- Puro, 1.5 μg of psPAX2, and 0.5 μg of pMD2.G was used to transfect cells using lipofectamine 3000 (Thermo Fisher) following the manufacturers’ protocols.

### 2.3. Western Blot and qRT-PCR

Cells were lyzed using RIPA buffer (Cell Signaling Technology, Boston, USA) at indicated time points. Twenty micrograms of total cellular lysate were loaded in each lane for SDS-PAGE gel electrophoresis. Separated proteins were transferred to the PDVF or NC membrane for Western blot analysis (Cytiva, Shanghai, China). Antibodies for β-actin (#4967), FLAG (#8146S), and MYC (#2276S) were purchased from Cell Signaling Technology. LPCAT3 (#ab232958) antibody was purchased from Abcam (Cambridge, UK). GAPDH (KC-5G4) antibody was purchased from KANGCHEN LLC. (Chengdu, China).

To quantify the mRNA expression, total RNA was extracted using TRIzol reagent (Takara), and the first-strand cDNA was generated using HiScript III RT SuperMix for qPCR (Vazyme Biotech) following the manufacturer’s instructions. The quantitative real-time PCR was performed on a Roche Light cycle 96 using ChamQ Universal SYBR qPCR Master Mix (Vazyme Biotech).

### 2.4. Data Process

Experiments were performed at least in three independent biological replicates and results were represented as mean ± SEM. Student’s *t*-test was used for statistical analyses.

## 3. Results

### 3.1. Exogenous SARS-CoV-2 Mpro Induced LPCAT3 Cleavage

After cell entry, the coronavirus releases genomic RNA into the cytoplasm [3], which encodes viral proteins, including the papain-like protease (PLpro) and the main protease (Mpro, or 3CLpro) [10]. Naturally, viral proteases cleave viral polyproteins pp1a/pp1ab to generate non-structural proteins (NSPs) [10]. Viral proteases also cleave host proteins, including IRF3, STATs, and ISGs [18,19,20].

In cellular lysate from SARS-CoV-2-infected HEK293T-hACE2 cells, an unexpected band (~37 kDa) was noticed when the LPCAT3 (~55 kDa) was analyzed, suggesting potential LPCAT3 cleavages during SARS-CoV-2 infection (Figure 1A). A slight decrease was observed for full-length LPCAT3 (Figure 1A). To test whether Mpro leads to the cleavage, a Flag-tag was fused to the C-terminus of SARS-CoV-2 Mpro in a pcDNA3.1 vector to generate pcDNA3.1-SCV2 Mpro (Figure 1B), since additional tags in the N-terminus of Mpro would dramatically reduce its catalytic activity. Interestingly, the cleaved LPCAT3 was clearly found with the expression of exogenous SARS-CoV-2 Mpro in HEK293T cells, indicating that Mpro was enough to induce LPCAT3 cleavage (Figure 1C).

Moreover, the LPCAT3 antibody used here was polyclonal. It was unclear whether the unexpected band was a truncated form of LPCAT3 or an off-target of the polyclonal antibody. To exclude the potential off-targets, a Myc-tag was fused to the N-terminus of LPCAT3 (Figure 1B). Notably, the truncated Myc-tagged LPCAT3 was observed when SARS-CoV-2 Mpro was expressed in cells (Figure 1D). This confirmed that exogenous SARS-CoV-2 Mpro could induce LPCAT3 cleavage. Afterward, increased LPCAT3 cleavage was found following the accumulation of exogenous SARS-CoV-2 Mpro in cells (Figure 1E), and the cleavage was dose-dependent on the expression level of Mpro (Figure 1F).

### 3.2. Mpro-Induced LPCAT3 Cleavage Might Be Related to the Gastrointestinal Symptoms in Coronavirus Diseases

SARS-CoV-2 infects various cell types causing different symptoms [5,34,35]. The crosstalk between viral proteases and host proteins is diverse in different cell types. To test whether SARS-CoV-2 Mpro would induce LPCAT3 cleavage in all types of cells, pcDNA3.1-SCV2 Mpro was used to transfect a series of cell lines, including human colorectal cancer-derived HCT116 cells (Figure 2A), human cervical cancer-derived HeLa cells (Figure 2B), human hepatocellular cancer-derived HuH-7 cells (Figure 2C), and human breast cancer-derived MCF-7 cells (Figure 2D). LPCAT3 expression was detected in all tested cells (Figure 2). As expected, cleaved LPCAT3 was found in all tested cells with exogenous SARS-CoV-2 Mpro expression, even though the expression level of Mpro was low in MCF-7 cells because of the poor transfection efficiency (Figure 2). It demonstrated that the SARS-CoV-2 Mpro-induced LPCAT3 cleavage was independent of cell types.

Most coronaviruses cause no or mild symptoms after infection exampling of HCoV-HKU1 and HCoV-OC43. However, some coronaviruses cause severe diseases, such as PEDV and MERS-CoV. LPCAT3 is abundant in the liver, intestines, and adipose tissue [26,28,36]. Since the important role of LPCAT3 in the intestine was mentioned earlier, it was hypothesized that Mpro-induced LPCAT3 cleavage might lead to gastrointestinal symptoms in coronavirus diseases. To test it, the Mpro of MERS-CoV, PEDV, HCoV-HKU1, and HCoV-OC43 was inserted into pcDNA3.1 vector with a Flag-tag at the C-terminus as SARS-CoV-2 Mpro, respectively. After verification by DNA sequencing, these recombinant plasmids were used to transfect HEK293T cells. Mpro of MERS-CoV and PEDV-induced LPCAT3 cleavage (Figure 3A,B). Even though the expression level of HCoV-HKU1 and HCoV-OC43 Mpro was relatively high, no LPCAT3 cleavage was observed (Figure 3C,D). It suggested that the Mpro-induced LPCAT3 might be related to the gastrointestinal symptoms of coronavirus diseases.

### 3.3. Mpro Indirectly Induced LPCAT3 Cleavage

The Mpro protein is highly conserved and cleaves similar motifs. However, the Mpro of SARS-CoV-2, MERS-CoV, and PEDV could induce LPCAT3 cleavage (Figure 1 and Figure 3), but the Mpro of HCoV-OC43 and HCoV-HKU1 could not (Figure 3). It hinted that the Mpro-induced cleavage of LPCAT3 might be indirect. GC376 was reported to covalently inhibit SARS-CoV-2 Mpro [37]. However, adding GC376 into the culture medium failed to abolish Mpro-induced LPCAT3 cleavage (Figure 4A). Afterwards, the proteasome is the major protein degradation machinery in eukaryotic cells [38]. MG132 was a promised reversible proteasome inhibitor, which had been proven to inhibit SARS-CoV-2 Mpro in our early studies [39]. Unexpected, Mpro-induced LPCAT3 cleavage was still observed in the presence of MG132 (Figure 4B), suggesting a proteasome-independent cleavage. Moreover, the 41st amino acid residue histone and the 145th amino acid residue cysteine of Mpro formed a catalytic dyad to hydrolyze substrates. To confirm that the catalytic activity of Mpro was not required for LPCAT3 cleavage, a Mpro mutant was constructed, in which the His41 was replaced by tyrosine and the Cys145 was replaced by serine as previously described [40]. Not surprisingly, Mpro mutation still induced LPCAT3 cleavage as well as wild-type SARS-CoV-2 Mpro (Figure 4C). Altogether, it clearly indicated that Mpro indirectly induced LPCAT3 cleavage.

### 3.4. Exogenous Mpro Caused ER Stress via LPCAT3 Cleavage

PEDV caused severe diarrhea in piglets, and notable gastrointestinal symptoms have been observed in 20–50% of patients infected with SARS-CoV-2 and MERS-CoV. It suggests that the gastrointestinal tract is an unignorable organ during coronaviral infection. LPCAT3 preferentially introduces polyunsaturated acyl onto the sn-2 position of lysophosphatidylcholine, modulating the membrane fluidity and playing an important role in lipoprotein production in the liver and intestine [26]. In addition, LPCAT3 knockdown in the liver exacerbated ER stress and inflammation [28]. Then, would Mpro-induced LPCAT3 cleavage induce ER stress in the intestine cells? Exogenous SARS-CoV-2 Mpro was expressed in HCT116 cells, which would induce LPCAT3 cleavage (Figure 2A). The C/EBP homologous protein (CHOP) expression was obviously higher in cells transfected with pcDNA3.1-SCV2 Mpro compared to cells transfected with empty vectors (Figure 5A). GRP78, another ER stress marker, was also increased in cells expressing exogenous SARS-CoV-2 Mpro (Figure 5B). Interestingly, HCoV-OC43 Mpro failed to induce LPCAT3 cleavage (Figure 3D), which did not boost the CHOP expression (Figure 5C). An increased CHOP expression was found when the LPCAT3 expression was knocked down by siRNA (Figure 5D,E). It demonstrated that Mpro-induced ER stress was related to LPCAT3 cleavage. Moreover, IPEC-J2 cells are porcine intestinal enterocytes, which are hard to transfect. A lentiviral vector encoding the PEDV Mpro was constructed to transduce IPEC-J2 cells (Figure 5F). PEDV Mpro was successfully and stably expressed in IPEC-J2 cells (Figure 5G) and increased CHOP mRNA was observed (Figure 5H).

## 4. Discussion

Coronaviruses infecting livestock and humans are zoonotic viruses. There is evidence indicating that HCoV-229E, -NL63, -OC43, and -HKU1 may originate from bats, rodents, cattle, and camels [41]. SARS-CoV and SARS-CoV-2 seem likely to have recently speciated from bat coronaviruses [42]. Considering the cross-species transmission history of coronaviruses, it is time to move the focus from human coronaviruses and take eyes on animal coronaviruses. Because of the phylogenetic conservation and the evolutionary history of coronaviruses, animal coronaviruses share many common features with human coronaviruses [43,44]. Many animal coronaviruses cause gastrointestinal symptoms. Although the respiratory tract symptoms caused by SARS-CoV-2 are a typical feature of COVID-19 [45], gastrointestinal symptoms have been observed in 20–50% of patients infected by SARS-CoV-2 [46]. Moreover, fatigue and gastrointestinal symptoms are typical features of long COVID [47], which may be related to the permanent damage in the gastrointestinal tracts by viral infection. Here, our studies indicated that Mpro indirectly cleaved LPCAT3, which was abundant in the intestine and essential for lipid absorption. Since malabsorption could lead to diarrhea, it suggested that the Mpro-induced LPCAT3 cleavage might play an important role in the gastrointestinal symptoms of coronavirus diseases.

LPCATs are essential to lipid metabolism and homeostasis. Based on their protein sequence, four LPCATs have been identified and classified into two families. LPCAT1 and LPCAT2 are members of the acylglycerophosphate acyltransferase family, which contains four conserved domains designated as LPA acyltransferase motifs. LPCAT3 and LPCAT4 belong to the membrane-bound O-acyltransferase (MBOAT) family, which contains MBOAT motifs but lacks the LPA acyltransferase motifs [26]. However, these LPCATs display distinct tissue distributions, enzymatic activities, and substrate preferences [48]. LPCAT3 is widely expressed and abundant in the testis, kidney, and metabolic tissues including the liver, intestine, and adipose [28,36]. In addition to the lysophosphatidylcholine (lysoPC) acyltransferase activity, LPCAT3 exhibits acyltransferase activities for lysophosphatidylethanolamine (lysoPE) and lysophosphatidylethanolamine (lysoPS) as substrates [49]. Most importantly, each LPCAT exhibits different acyl-CoA preferences. LPCAT3 prefers polyunsaturated fatty acyl CoAs (18:2-acyl-CoA or 20:4-acyl-CoA) as substrates [48]. Thus, LPCAT3 deficiency in the intestine reduces polyunsaturated phospholipid content and membrane fluidity, impairs passive fatty acid transport across the apical membrane of enterocytes, and decreases chylomicron assembly and secretion [27]. In this case, Mpro-induced LPCAT3 cleavage might lead to or aggravate gastrointestinal symptoms during viral infection. Since SARS-CoV-2, MERS-CoV, and PEDV Mpro could induce LPCAT3 cleavage but HCoV-HKU1 and HCoV-OC43 could not, it is reasonable to see diarrhea in patients and animals infected with SARS-CoV-2, MERS-CoV, and PEDV.

Recent studies have demonstrated that SARS-CoV-2 Mpro cleaves multiple host proteins, such as NF-κB essential modulator (NEMO) [21], Nod-like receptor family pyrin domain-containing protein 12 (NLRP12), transforming growth factor- β activated kinase 1 binding protein 1 (TAB1) [50], and signal transducer and activator of transcription 1 (STAT1) [19]. In this study, we have newly discovered SARS-CoV-2 Mpro-induced LPCAT3 cleavage. Unfortunately, the cleavage site and the mechanism are still elusive. There was a potent cleavage site in the N-terminus of human LPCAT3, which might be cleaved by Mpro. However, LPCAT3 might not be cleaved at that site since the cleaved fragment was too small (Figure 1). Moreover, LPCAT3 cleavage was independent of the catalytic activities of Mpro (Figure 4B). Scientists have put a lot of effort into the interaction between SARS-CoV-2 proteins and host proteins, which might be genetically associated with comorbidities of severe illness and long COVID [14]. Compared to other viral proteins, fewer host proteins have been identified to physically interact with proteases. Notably, we were convinced that SARS-CoV-2 Mpro and PLpro cleaved many host proteins, such as STAT1, TAB1, NLRP12, and NEMO. However, we failed to find them in these interactome studies. It is possible that the viral protease interacts and cleaves host proteins transiently, so it is difficult to identify associated proteins using physical interaction-based approaches, such as immunoprecipitation, pull-down assay, or yeast two-hybrid assay. Due to the elusive understanding of LPCAT3 cleavage during SARS-CoV-2 infection, it is still not clear what the biological impact of LPCAT3 cleavage is. Whether there are other host proteins involved in the cleavage of LPCAT3 by Mpro needs further investigation. Our preliminary results indicated that exogenous Mpro interrupted the lipid metabolism process in cells. However, the polyunsaturated phospholipid content and membrane fluidity of cells should be monitored in the future.

Importantly, diarrhea is the typical symptom of many animal coronavirus infections and leads to death. However, the antiviral agent remdesivir failed to alleviate diarrhea during PEDV and TGEV infection, even though these compounds efficiently inhibited viral replication. Our preprinting work indicated that a Mpro inhibitor stopped diarrhea in PEDV-infected piglets in 48 h, but viral genomic RNA was still detected until 7 days post-medicine management. This suggests that Mpro might directly cause diarrhea in coronavirus diseases, but the mechanism was still largely unclear. Our study suggested that Mpro-induced LPCAT3 cleavage might be critical to the gastrointestinal symptoms in coronavirus diseases, providing a hypothesis to explain how Mpro inhibitor could stop diarrhea before the clearance of viruses. Based on our studies, should inhibitors against LPCAT3 cleavage be screened for animal coronaviruses?

## Figures and Tables

**Figure 1 viruses-15-01696-f001:**
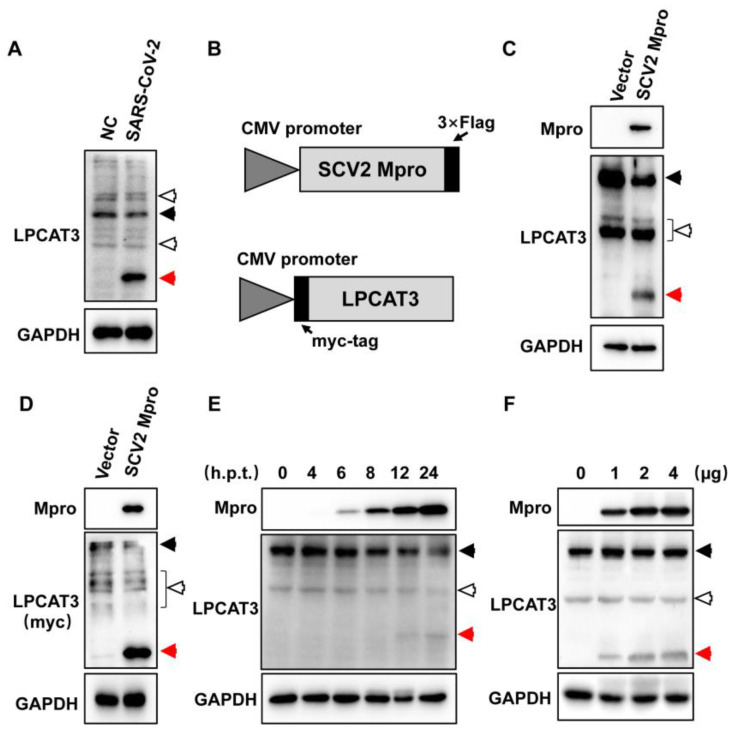
SARS-CoV-2 Mpro induced LPCAT3 cleavage. (**A**) Unexpected LPCAT3 fragments were found in SARS-CoV-2 infected cells. Twenty micrograms of cellular lysates were loaded in each lane and submitted to the Western blot analysis using anti-LPCAT3 and anti-GAPDH antibodies, respectively. NC, uninfected HEK293T-hACE2 cells. SARS-CoV-2, SARS-CoV-2-infected HEK293T-hACE2 cells. (**B**) Schematic of pcDNA3.1-SCV2 Mpro-Flag and pcDNA3.1-Myc-LPCAT3. (**C**) Exogenous SARS-CoV-2 Mpro in HEK293T cells induced LPCAT3 cleavage. HEK293T cells were transfected with pcDNA3.1 (Vector) and pcDNA3.1-SCV2 Mpro-Flag (SCV2 Mpro) for 24 h, respectively. Twenty micrograms of cellular lysates were loaded in each lane and submitted to Western blot analysis using anti-Flag, anti-LPCAT3, and anti-GAPDH antibodies, respectively. (**D**) SARS-CoV-2 Mpro induced exogenous LPCAT3 cleavage. HEK293T cells were transfected with pcDNA3.1-Myc-LPCAT3 recombination with either pcDNA3.1 (Vector) or pcDNA3.1-SCV2 Mpro-Flag (SCV2 Mpro) for 24 h. Western blot analysis was performed as in (**C**). (**E**) SARS-CoV-2 Mpro induced LPCAT3 cleavage at different time points post-transfection. HEK293T cells were transfected with pcDNA3.1 (0) or pcDNA3.1-SCV2 Mpro-Flag. The cellular lysate was harvested at time points as indicated in the figure. Western blot analysis was performed as in (**C**). (**F**) SARS-CoV-2 Mpro induced LPCAT3 dose-dependently. HEK293T cells were transfected with the indicated amount of pcDNA3.1-SCV2 Mpro-Flag for 24 h. Western blot analysis was performed as in (**C**). The black solid arrows present uncleaved-LPCAT3, the red arrows present cleaved-LPCAT3, and the black hollow arrows present non-specific bands.

**Figure 2 viruses-15-01696-f002:**
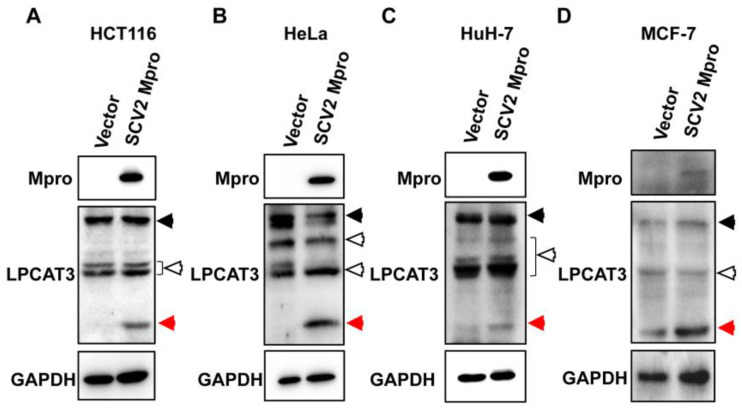
LPCAT3 cleavage induced by SARS-CoV-2 Mpro was found in most cell lines. Plasmids pcDNA3.1(vector) and pcDNA3.1-SCV2 Mpro-Flag (SCV2 Mpro) were used to transfect HCT116 (**A**), HeLa (**B**), HuH-7 (**C**), and MCF-7 cells (**D**), respectively. Cellular lysates were harvested and submitted to Western blot 24 h post-transfection. The black solid arrows present uncleaved-LPCAT3, the red arrows present cleaved-LPCAT3, and the black hollow arrows present non-specific bands.

**Figure 3 viruses-15-01696-f003:**
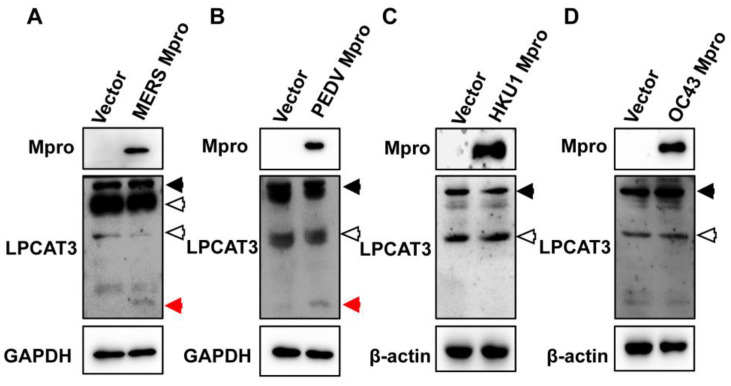
Mpro from highly virulent coronaviruses caused LPCAT3 cleavage. HEK293T cells were transfected with recombinant plasmids encoding various coronaviral Mpro, respectively. Cellular lysates were harvested and submitted to Western blot 24 h later. (**A**) MERS-CoV. (**B**) PEDV. (**C**) HCoV-HKU1. (**D**) HCoV-OC43. The black solid arrows present uncleaved-LPCAT3, the red arrows present cleaved-LPCAT3, and the black hollow arrows present non-specific bands.

**Figure 4 viruses-15-01696-f004:**
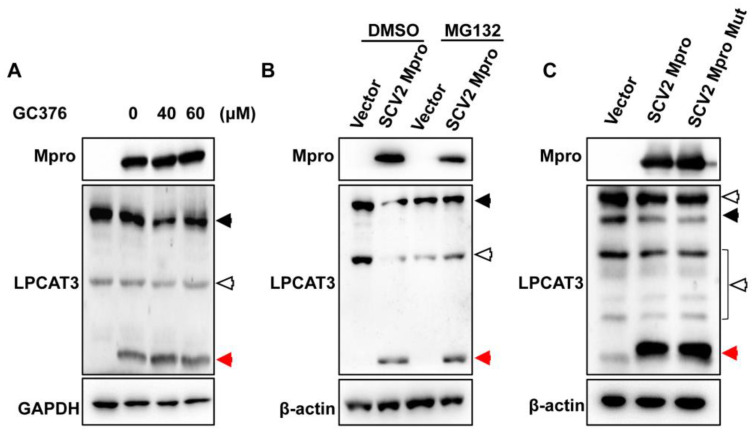
Mpro indirectly induced LPCAT3 cleavage. (**A**) SARS-CoV-2 Mpro inhibitor GC376 could not abolish Mpro-induced LPCAT3 cleavage. HEK293T cells were transfected with pcDNA3.1 (Vector) and pcDNA3.1-SCV2 Mpro-Flag (SCV2 Mpro). Overnight, cells were treated with GC376 at the final concentration as indicated in the figure for 12 h. Twenty micrograms of cellular lysates were loaded in each lane and submitted to Western blot analysis using anti-Flag, anti-LPCAT3, and anti-GAPDH antibodies. (**B**) Proteasome inhibitor MG132 could not abolish Mpro-induced LPCAT3 cleavage. HEK293T cells were transfected with pcDNA3.1 (Vector) and pcDNA3.1-SCV2 Mpro-Flag (SCV2 Mpro). Overnight, cells were treated with MG132 at the final concentration of 5 µM for 12 h. Twenty micrograms of cellular lysates were loaded in each lane and submitted to Western blot analysis using anti-Flag, anti-LPCAT3, and anti-GAPDH antibodies. (**C**) SARS-CoV-2 Mpro mutant could not abolish Mpro-induced LPCAT3 cleavage. HEK293T cells were transfected with either wild-type (WT) or protease-dead mutant (MUT) for 48 h. Twenty micrograms of cellular lysates were loaded in each lane and submitted to Western blot analysis using anti-Flag, anti-LPCAT3, and anti-GAPDH antibodies. The black solid arrows present uncleaved-LPCAT3, the red arrows present cleaved-LPCAT3, and the black hollow arrows present non-specific bands.

**Figure 5 viruses-15-01696-f005:**
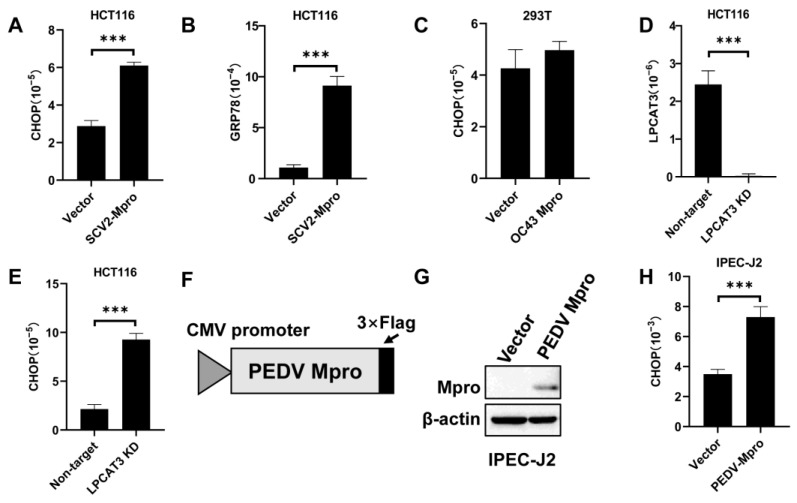
Mpro might induce ER stress via LPCAT3 cleavage. (**A**) SARS-CoV-2 Mpro induced CHOP expression. (**B**) SARS-CoV-2 Mpro induced GRP78 expression. HCT116 cells were transfected with pcDNA3.1 (vector) and pcDNA3.1-SCV2 Mpro-Flag (SCV2 Mpro) for 48 h. mRNA levels of CHOP (**A**) and GRP78 (**B**) were measured by qRT-PCR, and normalized to 18S rRNA. (**C**) HCoV-OC43 Mpro did not induce CHOP. Similar experiments were performed as in (**A**) but using HEK293T cells instead of HCT116. (**D**) LPCAT3 knockdown. (**E**) LPCAT3 knockdown induced the CHOP expression. HCT116 cells were transfected with non-target and LPCAT3-targeted siRNA for 48 h, respectively. mRNA levels of LPCAT3 (**D**) and CHOP (**E**) were measured by qRT-PCR, and normalized to 18S rRNA. (**F**) Schematic of pLVX-PEDV Mpro-Flag. (**G**) PDEV Mpro expression in IPEC-J2 cells. (**H**) PEDV Mpro induced CHOP expression. IPEC-J2 cells transduced by pLVX (Vector) and pLVX-PEDV Mpro-Flag (PEDV Mpro). Twenty micrograms of cellular lysates were loaded in each lane and submitted to Western blot analysis (**G**). mRNA level of CHOP was analyzed by qRT-PCR, and normalized to GAPDH (**H**). For all figures, experiments were performed at least in three independent biological replicates and data were represented as mean ± SEM. Statistical significance is from pooled data of the multiple independent experiments (*** *p* ≤ 0.001).

## Data Availability

Data are available upon request.

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
