# Peer review of "Coronaviral Main Protease Induces LPCAT3 Cleavage and Endoplasmic Reticulum (ER) Stress"

_viruses, 2023, doi:10.3390/v15081696_

Round 1

Reviewer 1 Report

The manuscript viruses-2498651, by Wang J et al., entitled "Coronaviral Main Protease Induces LPCAT3 Cleavage and ER Stress" requires several clarifications.

L16-24.- The meaning of all acronyms must be indicated when they appear for the first time. Sucha as, lysophosphatidylcholine acyltransferase 3 (LPCAT3), and so on.

L16-24.- The abstract is confusing. This must contain background, objective, material and methods, results and conclusions. The authors must rewrite it.

L68 -76.- This paragraph is inconsistent.

L77-80.- What is the hypothesis of this work?

L165, 184-186.- The authors infer that LPCAT3 has a single intestinal location, however LPCAT3 is located in the ER and is more abundantly expressed in the liver, pancreas, and adipose tissue.

·         Review: Zhao Y, et al. Identification and characterization of an important hepatic lysophosphatidylcholine acyltransferase. J Biol Chem. 2008;283(13):8258-65. doi: 10.1074/jbc.M710422200, and citations 19, 21 and 42.

In addition, the authors only present experiments with HCT116 and IPEC-J2 and do not compare them with cell lines from liver, pancreas, adipose tissue, and kidney.

L273.- What are the limitations of the work?

Reviewer 2 Report

Wang et al report on the cleavage of LPCAT3 induced by main proteases of some, but not all tested coronaviruses via a mechanism likely independent of Mpro catalysis, as shown by mutational studies. In general, investigations on the scope of coronaviral proteases with respect to host substrates are useful to inform on pathogenicity and are currently understudied. Hence, I support publication of this work after major corrections in viruses.

Major concerns:

My major concern is the unclear mechanism of Mpro induced cleavage and the insufficient characterization of the cleavage products, which should facilitate mechanistic studies. The identity of the cleavage products should be determined using mass spectrometry to identify the likely location of the LPCAT3 cleavage, so to potentially inform on the mechanism and on potential (host) proteases which could catalyse this reaction.

Please comment on whether the LPCAT3 sequence contains any obvious cleavage sites for mpro enzymes. What is the impact of the LPCAT3 cleavage on its catalytic activity?

What is the hypothesis why Mpro of SARS-CoV-2, MERS, and PEDV but not of OC43, HKU1 can cleave LPCAT3? Are there any hints based on fold or sequence of the Mpro enzymes?

To increase the relevance of the work, I strongly suggest to perform proteomic studies with coronavirus infected cells to investigate whether LPCAT3 cleavage is observed. It may be that this information is already publicly available in published proteomic data sets (see Nature Communications 2012, 12, 5553).

In general, the introduction and discussion should be more comprehensive with respect to host substrates of coronavirus proteases.

Figures 1-4; please label bands in Western blots accurately, what are the un-labelled bands; explain arrows in figure legends.

It is possible that Mpro induces ER stress, which triggers cleavage of LPCAT3 – has this hypothesis been investigated? If not, ER stress should be induced in a Mpro-independent manner to investigate effects on LPCAT3.

Minor concerns:

-          Line 16: specify exemplary coronaviruses

-          Line 36: specify reported gastrointestinal symptoms

-          Line 43: structural and accessory proteins

-          Line 51: cleaves ISG15ylated IRF3

-          Line 52: Mention Refs 44 and Nat. Commun. 13, 5285

-          Line 92 following: Specify strains, e.g. Wuhan, ..

-          Line 102: Plasmid sequences should be provided in SI

-          Line 130, see comment for line 51

   Line 22: better: suggesting the induction of ER stress in cells

 Line 23: ‘is a kind of’ be specific

-          Line 64-65: sentence unclear

-          Line 77: we report

-          Line 156: respectively

Round 2

Reviewer 1 Report

The authors have done an excellent job addressing all of my comments on the paper titled "Coronaviral Main Protease Induces LPCAT3 Cleavage and ER Stress." I have no further suggestions. I think the manuscript is acceptable for publication in "Viruses".